# Temporal dynamics of gut microbiomes in non-industrialized urban Amazonia

Ana Paula Schaan,[1,2,3] Amanda Vidal,[4] An-Ni Zhang,[5,6] Mathilde Poyet,[3,4,6,7,8,9] Eric J. Alm,[6,8] Mathieu Groussin,[2,3,5,6,8,9] Ândrea Ribeiro-dos-Santos[1,10]

**ABSTRACT** Increasing levels of industrialization have been associated with changes in gut microbiome structure and loss of features thought to be crucial for maintaining gut ecological balance. The stability of gut microbial communities over time within individuals seems to be largely affected by these changes but has been overlooked among transitioning populations from low- to middle-income countries. Here, we used metagenomic sequencing to characterize the temporal dynamics in gut microbiomes of 24 individuals living an urban non-industrialized lifestyle in the Brazilian Amazon. We further contextualized our data with 165 matching longitudinal samples from an urban industrialized and a rural non-industrialized population. We show that gut microbiome composition and diversity have greater variability over time among non-industrialized individuals when compared to industrialized counterparts and that taxa may present diverse temporal dynamics across human populations. Enterotype classifications show that community types are generally stable over time despite shifts in microbiome structure. Furthermore, by tracking genomes over time, we show that levels of bacterial population replacements are more frequent among Amazonian individuals and that non-synonymous variants accumulate in genes associated with degradation of host dietary polysaccharides. Taken together, our results suggest that the stability of gut microbiomes is influenced by levels of industrialization and that tracking microbial population dynamics is important to understand how the microbiome will adapt to these transitions.

**IMPORTANCE** The transition from a rural or non-industrialized lifestyle to urbanization and industrialization has been linked to changes in the structure and function of the human gut microbiome. Understanding how the gut microbiomes changes over time is crucial to define healthy states and to grasp how the gut microbiome interacts with the host environment. Here, we investigate the temporal dynamics of gut microbiomes from an urban and non-industrialized population in the Amazon, as well as metagenomic data sets from urban United States and rural Tanzania. We showed that healthy non-industrialized microbiomes experience greater compositional shifts over time compared to industrialized individuals. Furthermore, bacterial strain populations are more frequently replaced in non-industrialized microbiomes, and most non-synonymous mutations accumulate in genes associated with the degradation of host dietary components. This indicates that microbiome stability is affected by transitions to industrialization, and that strain tracking can elucidate the ecological dynamics behind such transitions.

**KEYWORDS** gut microbiome, temporal dynamics, industrialization, urbanization, Amazon, metagenomics

The gut microbiome consists of complex and dynamic communities of microbes that interact closely with host physiology and have a large role in maintaining human

Address correspondence to Ândrea Ribeiro-dos-Santos, akelyufpa@gmail.com.

Mathieu Groussin and Ândrea Ribeiro-dos-Santos contributed equally to this article.

E.J.A. is a co-founder and shareholder of Finch Therapeutics, a company that specializes in microbiome-targeted therapeutics.

See the funding table on p. 15.

health (1). These communities are sensitive to perturbations, and it is well established that the composition of gut microbiomes is dependent upon environmental forces, such as host diet, antibiotic use, and sanitation (2–4).

Existing research recognizes the critical role of lifestyle and subsistence strategies in determining the structure and function of the gut microbiome. Overall, individuals living in rural non-industrialized societies show gut microbiomes with higher levels of diversity and abundance of dietary fiber-degrading microbial taxa and genes compared to urban industrialized populations (4–7).

Among several features that distinguish microbiomes based on lifestyle, compositional shifts driven by annual seasonality have been presented as putative hallmarks of the most rural societies (8, 9). Such seasonal variability, reported among the Hadza foragers of Tanzania, is thought to be caused by the cyclical availability of certain food groups, which in turn drives the cyclical abundance of bacterial taxa and phylogenetic diversity (9). Since humans are increasingly transitioning from foraging lifestyles to industrialization, the absence of defining characteristics of rural microbiomes may indicate a loss of potentially fundamental properties of host-microbiome adaptations in urbanizing populations.

To understand how transitions to urbanized and industrialized lifestyles affect gut microbiome health, it is crucial to determine how these communities behave over time. Existing studies on the gut microbiomes of people living in urban areas report highly resilient and generally stable communities, despite fluctuations on a monthly or even daily scale (10–12). Defining what is expected in terms of temporal variations for healthy individuals will aid in the understanding of microbiome-associated disease development. For instance, tracking the diversity of gut microbiomes over time can be used to predict the onset of type 2 diabetes (13).

In low- and middle-income countries, urbanization of human populations is rapid and often accompanied by high levels of socio-economic disparities, nutritional shifts toward ultra-processed foods, and low-grade chronic inflammation (14), all of which are expected to have a drastic impact on the gut microbiome. However, what is currently known about the temporal dynamics of gut microbial populations is largely biased by data generated from high-income countries (12, 15–18).

Previously, we demonstrated the existence of an urbanization gradient in the Brazilian Amazon. Our results showed that the urban non-industrialized population of Belém, located on the margins of the Amazon basin in Brazil, shares gut microbiome signatures with local indigenous populations, while also having similarities with urban individuals from the United States and individuals under urbanization transition in Cameroon, which overall characterizes an urbanization transition (6).

Here, we investigate the presence of seasonality and temporal stability in the gut microbiome of individuals from Belém. The selection of time points aligns with the rainforest seasons experienced by the rural and non-industrialized (RN) population (9), facilitating comparisons with existing data. Our decision to capture distinct seasons in the Amazon region is also aimed at understanding whether there is an impact on the consumption of certain food groups, and whether the changing landscape of the urban non-industrialized Amazonian city during wet and dry seasons may influence the diversity and composition of gut microbiomes. By employing longitudinal shotgun metagenomic sequencing, we also examine this through species- and strain-level dynamics.

## RESULTS

### Lifestyle variation between Amazonian seasons

In order to investigate the temporal variation in gut microbiomes of urban Amazonia, we enrolled 24 adult individuals living in the urban non-industrialized (UN) Amazonian city of Belém, Brazil. We generated metagenomics shotgun sequencing data from two longitudinally sampled stool samples from each individual ($N = 48/24$ individual $\times$ two time points). To include both rainforest seasons, samples were collected in March/April

2021 (wet season, time point 1 [T1]) and July/August 2021 (dry season, time point 2 [T2]). Additionally, 3-day self-reported dietary records were collected for all individuals at both time points, along with other lifestyle metadata (Table S1).

To effectively explore how microbiome stability may be affected by levels of urbanization and industrialization, we included two data sets of gut microbiome longitudinal shotgun metagenomics from the literature in our analysis. To represent an urban industrialized (UI) population, we retrieved data from 127 samples from the Broad Institute-OpenBiome Microbiome Library (BIO-ML), which consist of longitudinally collected stool samples of individuals living in the Boston area, United States (12). To match our time series data from UN, we filtered data from UI to include individuals who had collected stool samples in the interval of 5 to 7 months, which yielded a total of 27 individuals (Fig. 1A).

As an RN population, we analyzed data from 38 stool samples of Hadza foragers living in Tanzania that were collected across seasons (9). Importantly, only three individuals from the RN population were longitudinally sampled and selected for shotgun sequencing. Nonetheless, we incorporated this data set as it consists of the only record of seasonal cycling of the gut microbiome in a rural population; when individual-level data were crucial (strain genomic profiling), the data set was disregarded from the analysis. For all populations, we hereby refer to time points as T1 (wet season or first collection date) and T2 (dry season or second collection date) (see Materials and Methods).

Analysis of dietary records from UN individuals showed that the average nutrient intake did not change between seasons in regards to animal proteins, total fats, fiber, and dairy consumption. However, we observed a decrease in the average intake of plant proteins and an increase in the consumption of açaí, an important local food staple, in T2 (Fig. S1). Overall, reported dietary habits were similar to what we had previously

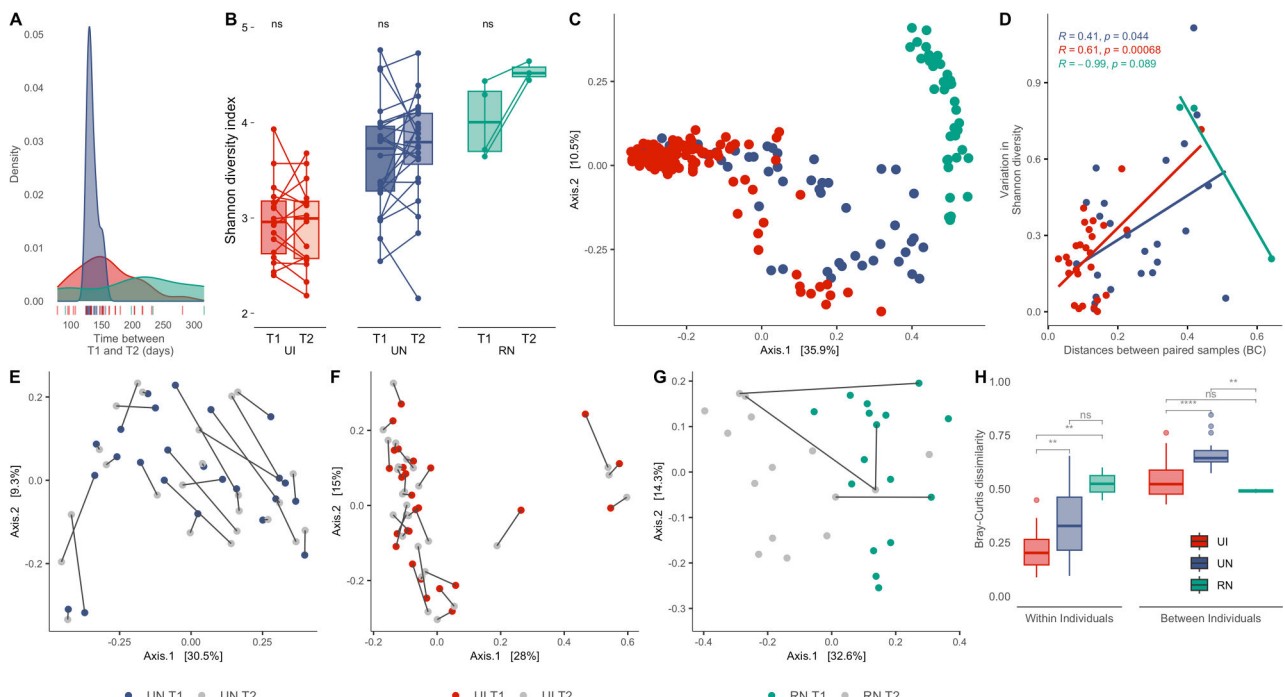

**FIG 1** Non-industrialized gut microbiomes have lower compositional stability. (A) Density plot showing the interval between sample collection for each population (in days). (B) Alpha diversity levels calculated using Shannon diversity index. Lines connect individuals across sampling time points. (C) PCoA plot of Bray-Curtis beta diversity distances comparing all three populations. (D) Scatter plot showing the association between Shannon diversity variation and Bray-Curtis distances between individuals. (E, F, G) Principal coordinate analysis of Bray-Curtis distances between samples from UN, UI, and RN, respectively, lines connect samples from the same individual. (H) Boxplots of Bray-Curtis dissimilarities within and between individuals. (ns, $P > 0.05$; *, $P \leq 0.05$; **, $P \leq 0.01$; ***, $P \leq 0.001$; ****, $P \leq 0.0001$).

described (6), consisting of daily intake of unprocessed, natural ingredients such as cassava flour and tropical fruits combined with ultra-processed food items.

## Stability of microbial diversity and composition across industrialization levels

To test whether non-industrialized populations harbor a less stable gut microbiome than industrialized individuals, our first set of analyses aimed to characterize how microbiome diversity is impacted across time (Fig. 1). As expected, overall alpha diversity measures followed a gradient of urbanization, with the lowest microbial richness in the UI microbiomes, followed by UN and RN (Shannon index, Kruskal-Wallis, $P = 1.4e - 11$) (Fig. S2). Seasonal-dependent variation in diversity was not observed in UN, given there is no trend in increase or decrease of microbiome richness for any given time point, unlike the RN individuals, who display increased levels of alpha diversity during T2 (Fig. 1B). It is possible, however, that this increase is primarily contributed by individual differences since only three individuals were included in this analysis (individuals with two longitudinal paired samples).

A principal coordinate analysis (PCoA) of Bray-Curtis distances between populations shows discrete population clustering with several points of overlap between UN and UI, which constitute the two urbanized groups in our analysis (Fig. 1C). Pearson correlation coefficients were used to test whether alpha diversity is associated with levels of compositional shifts in the gut microbiome. We found that increased values of alpha diversity variation between time points (UI, $R = 0.61$, $P = 0.00068$; UN, $R = 0.41$, $P = 0.044$) were positively associated with intra-individual Bray-Curtis distances over time for urbanized populations (Fig. 1D).

Furthermore, we observed signs of seasonal-based clustering among the RN samples, which was not the case for UN individuals (Fig. 1E through G). Nonetheless, in UN and UI, differences within individuals were greater than between individuals, illustrating the individual-specific nature of gut microbiome structure (Fig. 1H). Species-level analyses revealed that UI and UN individuals maintained a similar proportion of taxa from T1 to T2 ($0.84 \pm 0.12$ and $0.89 \pm 0.16$, respectively) (Wilcoxon, $P = 0.42$). Such inferences could not be drawn from the RN cohort due to the constrained longitudinal sampling size.

Next, we used analysis of composition of microbiomes (ANCOM) (19) to investigate whether such compositional differences could be driven by variability in the abundance of specific taxa between time points. At the species level, no taxa were found to be differentially abundant across time points for UN and UI individuals. For RN, however, ANCOM reported a total of 10 differentially abundant species between time points. For inter-population analysis, results showed a total of 110 bacterial genera to be differentially abundant between all three populations when disregarding specific time points (Table S2).

Additionally, we used intra-class correlation coefficients (ICC) to determine the level of stability for the most prevalent taxa (present in at least 50% of individuals of each population at any given time) across urban populations. This metric is calculated based on abundance variance within and between individuals for each species (see Materials and Methods). An ICC of 0 indicates no stability, while an ICC of 1 shows high species stability (where within-individual variance < between-individual variance) over time.

We observed that the gut microbiome of individuals from UI has overall greater stability than that of UN (Wilcoxon, $P < 2.2e-16$) (Fig. 2A). The median ICC score for species in UI individuals was 0.63, while in UN, the median ICC was 0.4 (Table S3). Interestingly, we observed a positive correlation between intra-individual variance values for taxa present in both populations, which indicates these species have similar ecological dynamics among urbanized populations, regardless of abundance (Pearson's correlation, $R = 0.65$, Wilcoxon $P < 2.2e-16$, Bonferroni correction $P < 2e-16$) (Fig. 2B).

In spite of this comparable trend, however, gut microbes showed higher intra-individual variance among UN samples when compared to UI (Fig. 2C and D). *Enterobacter cloacae*, *Klebsiella pneumoniae,* and *Akkermansia muciniphila* showed the highest intra-individual variance and, therefore, the lowest stability in UN (Fig. 2C). On the other

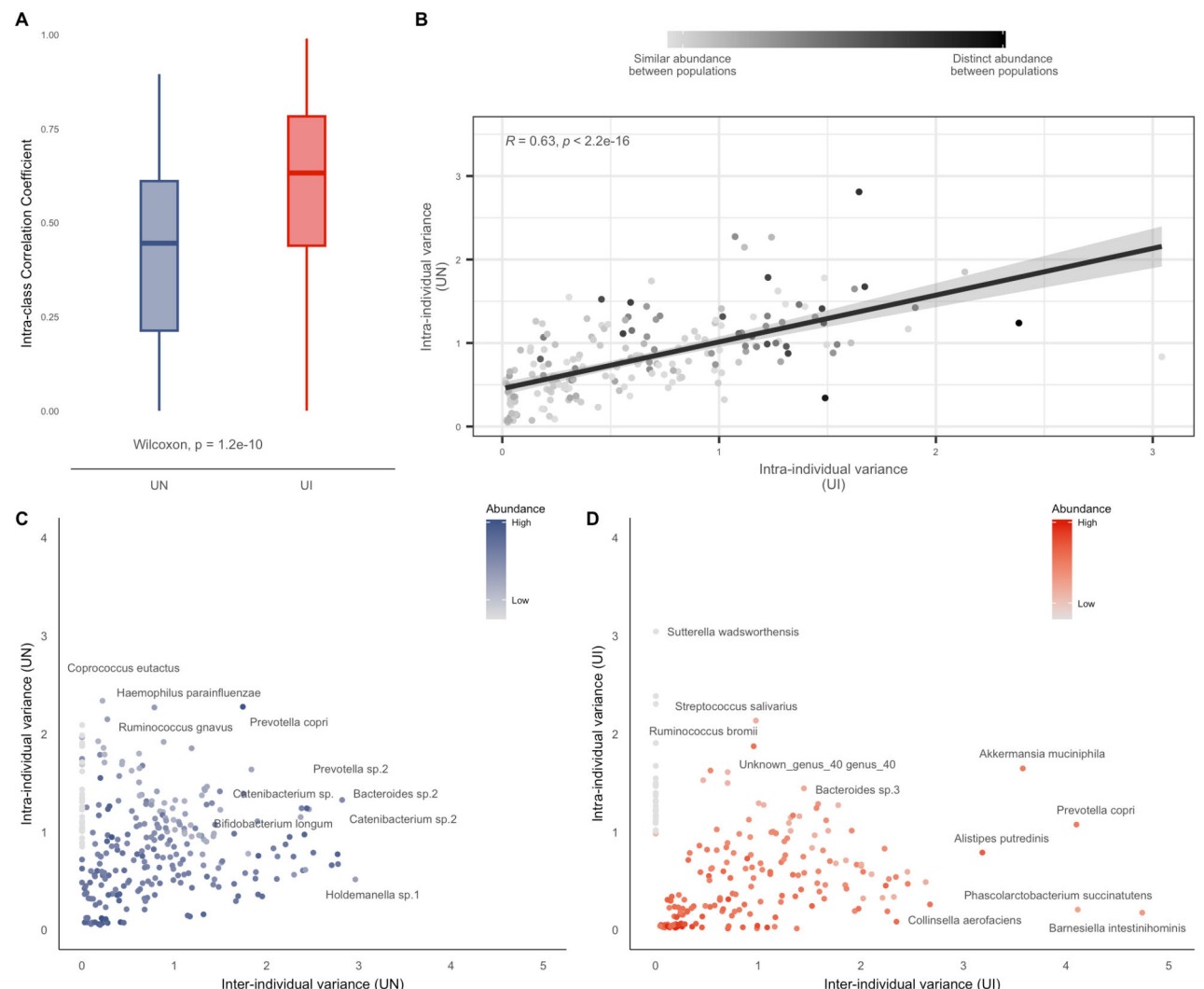

**FIG 2** UN individuals have higher intra-individual variance over time. (A) Boxplot showing the ICC for gut microbial species of UN and UI cohorts. (B) Scatter plot of intra-individual variance among species from UI and UN individuals. (C, D) Relationship between within- and between-individual species variance, as calculated using mixed-effects models. Species are colored according to their relative abundance in each respective population.

hand, for the UI individuals, the intra-individual variance was mostly driven by probiotic species such as *Streptococcus thermophilus* and *Streptococcus salivarius* (Fig. 2D).

Furthermore, *Prevotella copri* and *Prevotella* sp., which are generally associated with non-industrialized lifestyles or higher intake of dietary non-digestible fibers, showed distinct patterns of contribution to variance in both populations. Previously, high levels of within-individual abundance variability of *Prevotella copri* were associated with healthy individuals, while patients with inflammatory bowel diseases showed increased stability of this taxa over time (16). In our UN cohort, *P. copri* was highly abundant and more associated with fluctuation in abundance at the intra-individual level, whereas the opposite was observed for UI.

To assess whether temporal variations in microbiome composition result in changes in functional and metabolic potentials, we performed an alignment of protein-coding sequences predicted from contigs to the Kyoto Encyclopedia of Genes and Genomes (KEGG) Orthology database (Fig. S3). Using ANCOM, we tested the differential abundance of levels 2 and 3 KEGG pathways. The RN population was the only group to show significant differential abundances of metabolic pathways across sampling time points, as shown in Fig. S4. We did not observe any significant differences in the functional

potential of microbial communities across time points for the urban populations of UN and UI, which demonstrates a high level of functional stability and functional redundancy of the gut microbiome, even when subject to structural modifications over time (Fig. S3).

## Enterotype changes across time points

Classification of microbiomes based on enterotypes has been used to distinguish healthy and disease-prone gut communities (20). Here, we used community typing to test how three populations living in diverse levels of industrialization would be classified and grouped according to community composition. Additionally, our goal was to understand to what extent levels of microbiome stability affect enterotype classification over time. We employed a Dirichlet multinomial mixtures (DMM) model on abundance data for all three populations in our study and chose the optimal number of clusters based on the best model fit according to performance metrics, as shown in Fig. 3A.

Using samples from RN, UN, and UI, clustering based on community composition determined the existence of four distinct enterotypes (Fig. 3B and C). Two of these community types, hereby referred to as BactA and BactB, are driven by high abundances of the *Bacteroides* genus, as well as the presence of *Parabacteroides* and *Alistipes* taxa, respectively (Fig. 3B; Fig. S5A). The second set of enterotypes, PrevA and PrevB, is defined

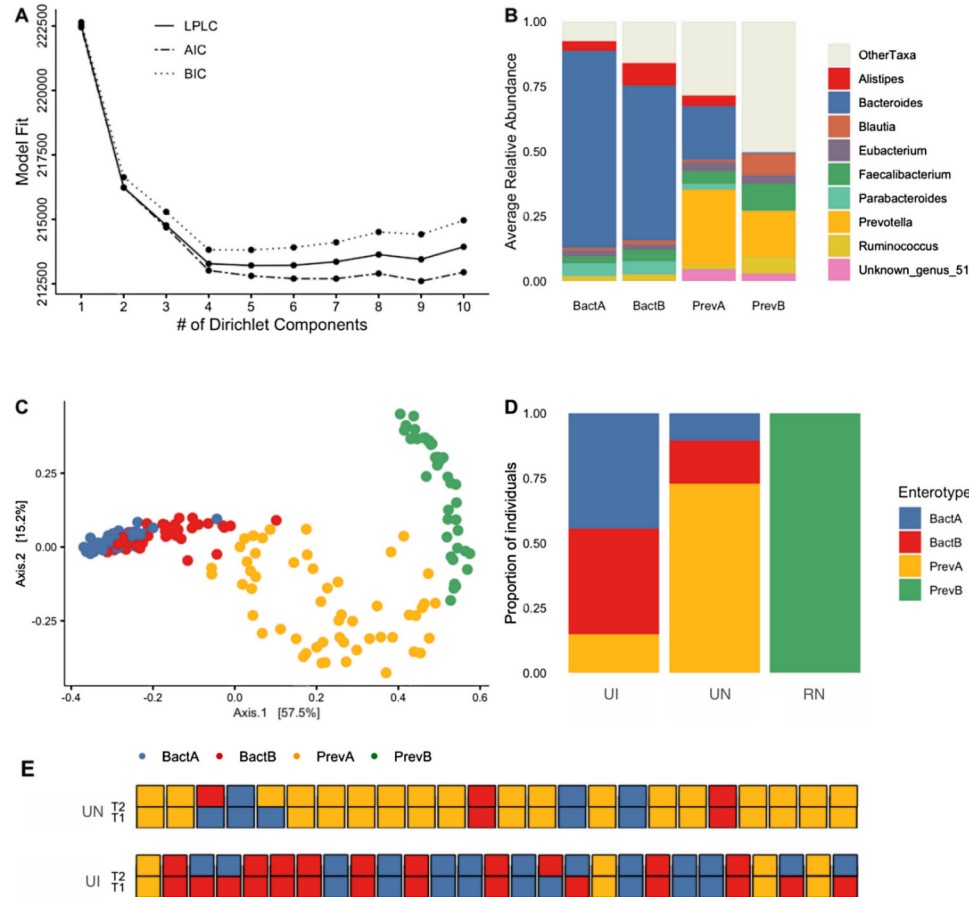

**FIG 3** Individual enterotypes are generally stable despite compositional shifts. (A) Optimal number of enterotypes (or community clusters) according to model performance metrics (LPLC, Laplace; AIC, Akaike information criterion; BIC, Bayesian information criterion). (B) Average relative abundance of driver taxa that define each enterotype. (C) PCoA of Bray-Curtis distances where colors indicate the enterotype of each sample. (D) Proportion of individuals per population that have gut microbiomes belonging to each enterotype classification. (E) Enterotype switches for each individual from UN and UI according to time point.

by high abundances of the *Prevotella* genus, with the marked distinction that PrevA includes increased abundances of *Bacteroides*, whereas PrevB has *Faecalibacterium* and *Blautia* as secondary drivers (Fig. 3B).

At the population level, we observed that the PrevB enterotype is exclusive to the RN individuals, which harbor a highly distinct microbiome composition from the other populations (Fig. 3D). Most samples from UN cluster into the PrevA enterotype, followed by BactB and BactA. On the other hand, BactA is the most common community type among UI samples, followed by BactB and PrevA.

A total of six individuals from the UI cohort changed enterotypes between time points, all of which transitioned from BactA to BactB or vice versa (Fig. 3E). In UN, this shift was seen in two individuals, which followed a gradient directionality as well, from BactA to BactB and PrevA to BactA. The compositional similarity of shifting enterotypes indicates that there are no drastic disruptions in community structure between time points for the two cohorts, indicating the stability of gut microbiomes over time in these populations.

In view of previous associations between *Bacteroides*-dominated enterotypes and systemic low-grade inflammation, diabetes, and obesity (20), we assessed body mass index (BMI) data for UN individuals to identify associations between increased BMI and community type. Our analysis showed that there was no significant association between BactA and BactB enterotypes and high BMI values (Kruskal-Wallis, $P = 0.11$) (Fig. S5B).

## Strain replacements as a genomic metric of microbiome stability

Population dynamics within different species in the gut microbiome may be influenced by strain-level interactions and replacements over time (12, 15). To understand microbial population stability in the gut microbiome of individuals living in distinct levels of industrialization, we investigated such evolutionary dynamics by recovering and analyzing genomes of individual strains across time points.

First, genomes were *de novo* assembled from metagenomic sequences. Overall, metagenome-assembled genomes (MAGs) had high quality (89% median completeness, 0.7% median contamination, a median length of 2,236,137 base pairs, and a median $N50$ of 30.513 base pairs), according to CheckM analyses (21). Outlier MAGs for contamination and completeness were removed from the analysis (see Materials and Methods). Next, we joined bins from each individual from both time points and used dRep v.3.2.2 (22) to dereplicate genomes.

In order to determine whether bacterial strains and populations were replaced over time, we identified strains that colonized the same individual at both time points and searched for single nucleotide polymorphisms (SNPs) and overall genomic identity (average nucleotide identity [ANI]).

To do so, we employed InStrain (23), which performs highly accurate genome comparisons on metagenomic data, on 22 pairs of individuals per population (Fig. S6A). We obtained a total of 506 pairwise genome comparisons, 254 from UN and 251 from UI individuals, spanning 35 distinct bacterial families, 83 genera, and 168 species (Table S4). Predictably, we found that shared strains were more common within individuals than between individuals, as shown in Fig. 4.

InStrain genome analyses yielded information for both strain replacements and population replacements (see "Strain profiling" in Materials and Methods) (Fig. S6B). In short, we called a strain replacement when two strains had an ANI of less than 0.99999. Population replacements were called in cases where ANI > 0.99999 when considering only sites with no shared alleles between strains.

While the frequency of strain replacements was not significantly different between urbanized populations, we observed that bacterial population replacements were almost twice as common in the gut microbiome of UN individuals ($\chi^2$ goodness-of-fit test, $P$-value <0.001). We further observed these events were most common among the *Bacteroides* genera (*Bacteroides* and *Bacteroides_B* account for 25% of bacterial population replacements in UN samples and 20% in UI samples) (Fig. 5A and B).

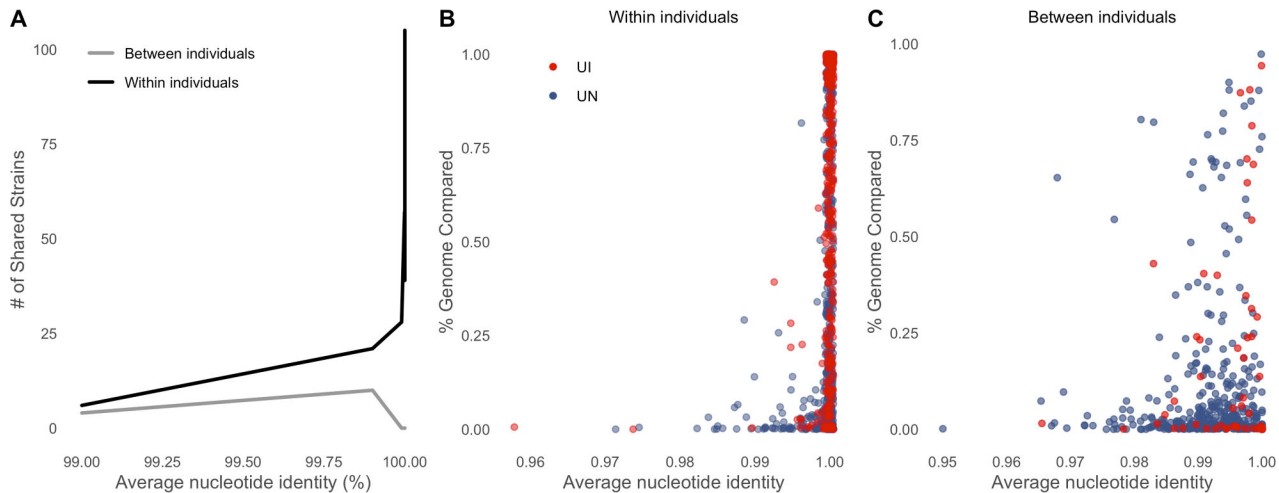

**FIG 4** Shared strains are more common within individuals than between individuals. (A) Shared strains within and between individuals, calculated with genome pairs in InStrain using 99.999% consensus average nucleotide identity (conANI) as a threshold to identify identical strains. (B, C) ANI of strain pairs considering the amount of overlap between genomes within and between individuals, respectively.

To identify which metabolic pathways may be affected by such ecological dynamics within the gut microbial community, we performed functional annotation on genes with detected SNPs belonging to replaced strains. As previously described (15), we focused on patterns of polymorphisms as a strategy to infer genotypes and subset sites that could

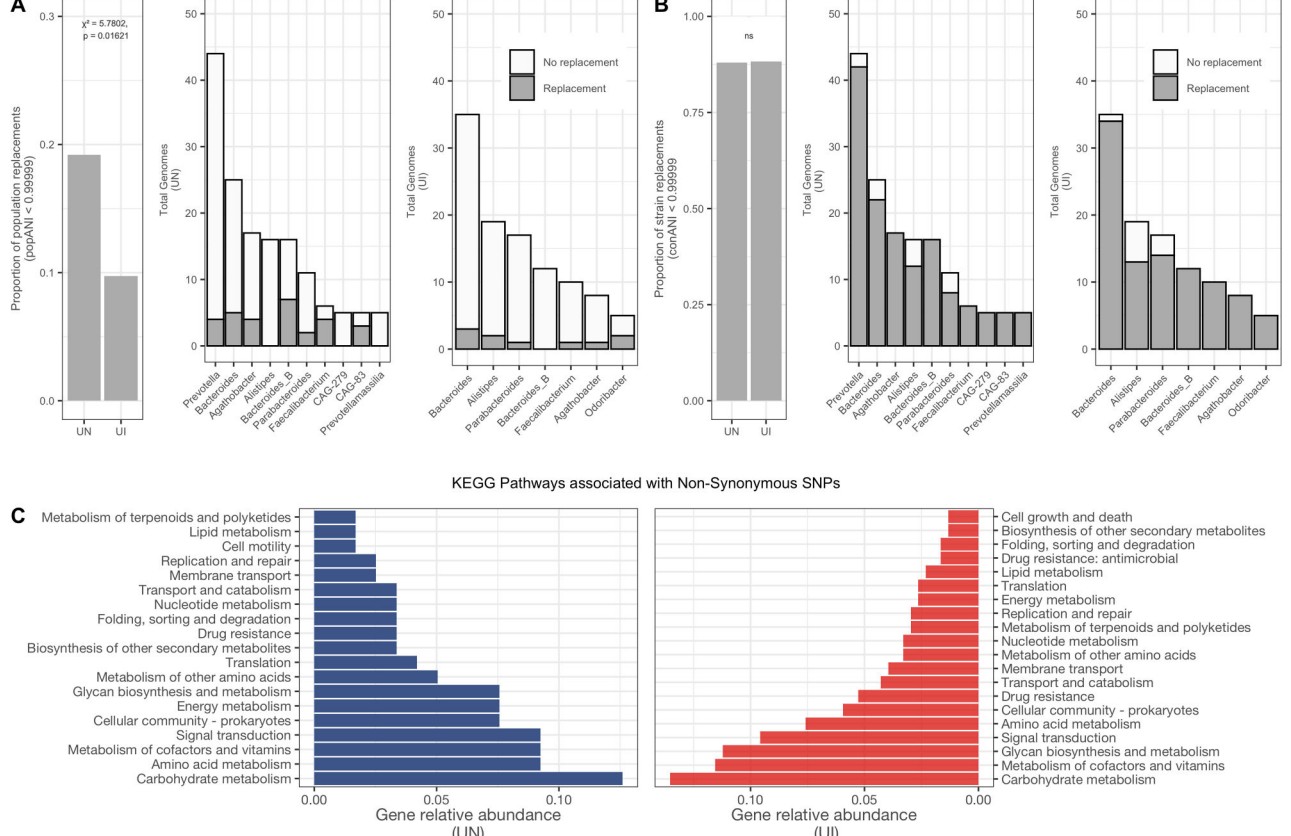

**FIG 5** Strain population replacements are more common in UN individuals. (A) Proportion of population average nucleotide identity (popANI) and their distribution across genera represented within MAGs. (B) Proportion of strain consensus average nucleotide identity (conANI) and their distribution across genera represented within MAGs. (C) Functional annotation of genes with detected non-synonymous SNPs from replaced strains.

confidently indicate true strain replacements (see "Strain profiling" in Materials and Methods) (Fig. S6C).

Our results showed that the majority of non-synonymous SNPs were located within genes involved in the metabolism of host dietary polysaccharides and functions likely to impact bacterial fitness within the gut, such as carbohydrate metabolism, amino acid metabolism, and glycan biosynthesis (Fig. 5C; Table S5). This pattern was shared by both urbanized populations despite their distinct industrialization levels. Finally, we did not observe differences in the abundance of affected metabolic pathways when analyzing individual species groups, indicating that these genomic functions are likely selected at the community and ecosystem levels.

## DISCUSSION

In this paper, we report the temporal dynamics of gut microbiomes from a non-industrialized urban population of the Brazilian Amazon. We placed our studied population in context by incorporating available data sets from individuals from divergent lifestyles but similar sampling time points intervals, in order to obtain an appropriate understanding of gut microbial variability across diverse populations.

Similar to what was reported for RN (9), we identified a slight shift in dietary intake among UN individuals. Previous studies had reported that changes in host diet are the main factor causing compositional shifts in the microbiome (2, 24). Here, we observed an increase in the consumption of açaí during T2. However, it is noteworthy that in T1, where açaí consumption is comparatively lower (almost zero for most individuals), the microbiome compositions between UN and UI still exhibit significant divergence. This suggests that while açaí consumption may contribute to some level of microbiome variations, it is not the sole driving force behind the observed differences in community composition.

Considering the UN population is located at the center of the urbanization gradient between the UI and RN individuals (according to alpha and beta diversity metrics), we assume that increased compositional differences across time points among non-industrialized populations may be linked to levels of transition to industrialization. This is further corroborated by our ICC results, which demonstrate that the gut microbiome of industrialized individuals has higher temporal stability than non-industrialized individuals. However, the positive correlation of intra-individual variance values among UN and UI individuals is likely an indication that species within the gut microbiomes of urbanized populations have comparable ecological dynamics.

Previous studies (10, 25) showed that abundance variability is part of the equilibrium dynamics of the healthy gut microbiome and that deviations are expected to occur within weekly or even daily timescales, eventually returning to a steady community state. Interestingly, the high intra-individual abundance variances in UI were driven by low abundance species typically associated with industrialized populations, namely *Enterobacter cloacae, Klebsiella pneumoniae*, and *Akkermansia muciniphila*, along with rural lifestyle-associated high abundance *Prevotella copri*. While these findings may indicate expected levels of abundance fluctuation, the species experiencing such deviations suggest that the microbial populations of UN gut microbiomes are currently adapting to an increasingly urbanized lifestyle. Thus, abundance variance may also be driven by external forces, such as shifts in host diet, causing high within-community abundance fluctuations.

Nevertheless, the unchanged functional profiles of gut microbiomes in our UN data set and evidence that individuals are more similar to themselves than to others further corroborate previous findings that the gut microbiome has evolved to maintain equilibrium for long periods of time (15, 17, 26–28). The significant differences we found in the functional profiles among the gut microbiomes of RN across were a confirmation of previously reported findings. This is suggested to be due to highly contrasting dietary patterns in the wet and dry seasons, which support the enrichment of functionally adapted bacterial taxa and consequently an alteration in metabolic profile (9). However,

due to the lack of same-individual longitudinal sampling in the RN cohort, these findings could be driven by individual-specific differences.

Our enterotyping analyses provide further validation of the extent of gut microbial stability. Populations from both the industrialized and non-industrialized cohorts rarely shifted to another enterotype in the course of 4 to 6 months, and diverging cases mostly transitioned to the next enterotype following a gradient-like shift. This also demonstrates that subtle compositional variations are enough to cause a switch of community type, thus supporting earlier studies that argue in favor of a non-discrete interpretation of such community profiles (29).

Genomic analyses showed that bacterial strain replacements, measured using consensus average nucleotide identity (conANI) were equally common among both urbanized populations, but bacterial population replacements, measured using population average nucleotide identity (popANI) were more frequent among UN individuals. In other words, in UN microbiomes, the genomic distances between former (T1) and newly (T2) fixed strains were higher than those in UI individuals. Furthermore, we observed that strain and population replacements were substantially driven by species within the *Bacteroides* genus. The fact that other genera did not show the same level of divergence and replacements suggests that temporal dynamics in the gut affect taxa in different ways (26, 30, 31).

One hypothesis is that UN individuals are exposed to higher microbial diversity (from environment and population-derived strains), which allows for the colonization of divergent strains over time and the consequent displacement of former commensals. Alternatively, it is possible that taxa highly associated with industrialized populations, such as *Bacteroides* (6, 24), experience increased ecological pressures to stably colonize the gut of transitioning non-industrialized individuals. Experimental work has demonstrated that within-host adaptive evolution is a common feature of gut commensals, even in the absence of disruptive perturbations such as antibiotic intake (32). Furthermore, previous data showed that evolutionary mechanisms, such as horizontal gene transfer, may play an important role in the adaptation of industrialized and transitioning microbiomes (7).

Another possibility is the existence of multiple co-occurring strains within the gut community of UN that were not fully captured at a given time point. According to earlier findings (12, 15), the coexistence of related strains within the gut microbiome may be the result of independent niche occupations, thus allowing for the stable presence of two or more members of the same phylogenetic clade. Nonetheless, our data suggest that increased strain replacements, and possibly strain diversity, may be features of non-industrialized and transitioning microbiomes.

Finally, our functional annotations of genes harboring non-synonymous SNPs across time points show that the metabolic pathways being selected across gut microbiomes are similar between both urbanized populations and mostly affect genes associated with the degradation of host dietary components. This suggests that evolutionary pressures acting upon gut microbiomes of both industrialized and non-industrialized individuals select functions that favor microbial utilization of host-derived carbohydrates, which has also been shown previously within gut genomes of *Bacteroides fragilis* (32) and is thought to drive the seasonal cycling of RN microbiomes (9).

This phenomenon may constitute an evolutionary adaptation that impacts microbial fitness by enhancing nutrient acquisition, essential within the competitive environment of the human gut (32). Moreover, increased capacity for polysaccharide degradation is likely to have important consequences for host metabolism and health, influencing the production of short-chain fatty acids and improving gut and metabolic homeostasis (33). Finally, these findings emphasize the adaptability and metabolic flexibility of gut microbiomes under diverse human industrialization levels.

A note of caution is due, considering that one limitation of our study was that the genomic divergence of strains was tracked only in cases where genomes for the same species were reconstructed across both time points. Given the nature of metagenomic

data, this means we were not able to identify the loss or confirm a decrease in the abundance of specific strains across time points.

Additionally, despite showing signals of seasonal cycling to some extent, the lack of longitudinal data derived from the shotgun metagenomic sequencing of stool samples of RN individuals, carried out by Smits et al. (9), is a significant limitation that hindered appropriate data analysis and impacted our ability to draw conclusions from individual-level data from two non-industrialized populations. Sample sizes are also a limitation within the UN cohort, for which we collected stool from 24 individuals (power = 0.4). It is crucial to acknowledge that this may impact the generalizability of our findings and may have a limited ability to detect small effects. Furthermore, as with any analysis of public metagenomic data, it is important to consider the possibility of batch effects emerging from different methods of sample collection and processing.

Taken together, our results show that gut bacterial communities of individuals living in urban environments are subject to distinct population dynamics and that levels of industrialization may have a role in determining their stability. Characterizing the temporal variability of gut microbiomes of Brazilian Amazonians and other non-industrialized populations is a fundamental step toward understanding the effects of industrialization on gut communities and overall host health. To fully grasp the extent and impact of strain replacements and compositional shifts, future research should focus on additional time series associated with the culture and isolation of bacterial strains to uncover both the short-term and long-term stability of non-industrialized gut microbiomes.

## MATERIAL AND METHODS

### Study cohort and sampling

We recruited a total of 24 individuals living in the city of Belém, Pará, Brazil. The average age of participants was 30 years old (18–59), and we recruited both male ($N = 9$) and female ($N = 15$) individuals. Participants recruited and included in this study did not report antibiotic medication intake at least 3 months prior to sample collection.

Belém is the capital and largest city of the Pará state, in the northern region of Brazil, and is located at the mouth of the Amazon River. It is classified as an urban non-industrialized city, according to criteria employed by the Global Microbiome Conservancy (7), due to a human development index of 0.689 and population density of approximately 3,800 inhabitants per square kilometer (34, 35). We aimed to recruit inhabitants of different regions of the city, which consist of medium-income urbanized centers and low-income stilt slum communities.

Participants were enrolled at the Federal University of Pará, Brazil, and we aimed to include individuals living in multiple regions of the city of Belém, to ensure adequate representation of the locations within the urban area. Participants were asked to provide stool samples at two time points during the year, the first in March/April of 2021, during the wet/rainy season of the Amazon, which spans from December through May. The second sample collection took place in July/August 2021, during the dry season, which occurs from June to November. For the purposes of this study, the wet/rainy season is referred to as time point 1 (T1) and the dry season is referred to as time point 2 (T2).

Stool samples were collected in sterile containers and returned to the research staff up to 6 hours after collection. Samples were stored in RNAlater stabilizing solution (Thermo Fischer Scientific) and frozen at −80°C until DNA extraction in the Laboratory for Human and Medical Genetics of the Federal University of Pará, Belém, Brazil.

### Literature data

Longitudinal gut microbiome metagenomic data were also included for two other populations in order to create an urbanization gradient for the purposes of our analyses. A total of 38 samples (from 34 individuals) from the RN individuals, representing the subset of samples submitted for shotgun sequencing from Smits et al. (9), were processed and analyzed in this study. The Hadza represent a rural non-industrialized

lifestyle. Importantly, only three individuals from this population were longitudinally sampled across at least two distinct seasons and were represented in shotgun metagenomic sequencing (namely, individuals 54, 55, and 56). Thus, RN samples were removed from genomic analyses that aimed to identify SNPs among microbial strains across time points and that required paired sequencing data.

A total of 127 longitudinal gut microbiome samples from the BIO-ML (12), collected in the Boston area, USA, were included to represent an urban and industrialized lifestyle. Of these, 27 individuals were sampled in periods ranging from 5 to 7 months apart and were selected for analysis of bacterial strain replacements in order to match sampling periods from UN. In the paper, this population is referred to as "UI."

We also classified RN samples into T1 and T2, where samples collected in early or late-wet periods (November to April) are referred to as T1, and those collected during early or late-dry periods (May to October) are referred to as T2. For UI samples, first collection date is referred to as T1, and second collection date as T2. Importantly, UI samples do not have seasonal intervals (i.e., winter/summer, spring/fall) in sampling collection. This information is relevant for identifying trends in alpha diversity measurements across time points. Thus, for this specific analysis, we included only individuals whose T1 sample was collected in either in fall/winter and whose T2 was collected in either spring/summer.

## Metadata collection

Dietary information was collected using food journals during 3 days prior to sample collection. This consisted of detailed information on meals and food consumption, which were later assessed for nutrient intake. We also applied a questionnaire for participants to provide information on previous and current clinically diagnosed diseases, medication intake, BMI, household type (brick, wooden, apartment building), sewage treatment or lack thereof, source of drinking water, pets, and the number of co-inhabiting individuals. All metadata were collected by a single research member.

## DNA extraction and sequencing

Total DNA from fecal samples was extracted using the DNeasy PowerSoil Kit (QIAGEN, Hilden, Germany) according to the manufacturer's protocol. Extraction was performed in a laminar flow hood and included negative controls to identify contamination from reagents or the environment. Eluted DNA was quantified with fluorometry and subsequently stored at −20°C. Library preparation for sequencing was carried out according to the Illumina DNA Prep protocol. Libraries were subsequently pooled and quantified with TapeStation (Agilent, Santa Clara, CA) before paired-end (2 × 150 bp) sequencing on the Illumina NextSeq500 platform.

## Metagenome quality control and processing

Sequencing yielded a median of 1.7 million reads per sample. Read quality was assessed with FastQC (36) and quality-filtered using FastUniq v1.1 (37) to dereplicate reads and Trimmomatic v0.39 (38) with parameters LEADING:3 TRAILING:3 SLIDINGWINDOW:5:20 MINLEN:50. We indexed the human genome (version hg 38) using Burrows-Wheeler Aligner (BWA) (39) and removed reads that aligned to human sequences using Samtools v1.9 (40).

## Taxonomic classification and diversity assessment

Filtered reads were assigned taxonomic labels with Kraken2 v2.0.8 (41) and abundance estimates were carried out by Bracken v2.5 (42). Taxonomic classification was based on sequences from the Kraken2 standard database completed with a sequence database derived from the Global Microbiome Conservancy isolate library (7, 12). Employing this custom database increased taxonomic profiling by 1.7 times when compared to

classification using solely the Kraken2 standard database (Wilcoxon, $P$ = <2.2e−16) (Fig. S6).

Alpha and beta diversity analyses were performed in R v.3.5.3 (43) using functions implemented in "phyloseq" and "vegan" packages (44, 45). For alpha diversity metrics, samples were rarefied to the minimum number of reads and calculated based on number of observed species, Shannon index, Chao1 index, and Simpson index. For beta diversity analyses, data were filtered to remove taxa not present in at least 10% of samples. Compositional dissimilarities were computed using relative abundances and genus-level taxonomy with Bray-Curtis distances.

## Intra-class correlation coefficient analysis

We employed ICC to determine the reliability of stability data for all 440 individual species across UN and UI individuals. First, the mean and variance of each species were calculated and fit into mixed-effects models using the lme4 package implemented in R (46). For this, we used $log_{10}$-transformed relative abundance counts for each species and fit the model by considering individuals are random variables and time points as fixed effects. The model outputs information regarding total variance, which is separated into within-individual ($\sigma^2_{WI}$) and between-individual variance ($\sigma^2_{BI}$). Subsequently, the ICC is calculated from this variance partitioning as follows: $\sigma^2_{BI} / (\sigma^2_{BI} + \sigma^2_{WI})$.

## Differential abundance analysis

In order to assess differential abundances between pairs of individuals and groups of populations, we used ANCOM (19), which compared absolute abundances of taxa while accounting for the compositional nature of microbiome data. Here, we considered taxa as differentially abundant if they passed the W-statistic 0.6 cutoff.

## Enterotype assignments

Enterotyping (also known as community typing) was performed on genus-level relative abundance data filtered for amplicon sequence variants (ASVs) present in at least 50% of samples using DMM models, as described in Holmes et al. (47). This method uses probabilistic modeling to calculate the optimal number of clusters (enterotypes) and classify each sample into an enterotype based on compositional similarity. These calculations were applied to our data set including all 213 samples from all three populations using the DirichletMultinomial package version 1.24.1 implemented in R (43, 47). Measures of model performance (Akaike information criterion, Bayesian information criterion, Laplace) chose the model that best describes the observed data.

## Metagenome assembly

Samples were processed independently for *de novo* genome assembly with metaSPAdes software, implemented in SPAdes v3.15.3 (48, 49). Contigs were indexed, aligned, and sorted using Samtools v1.9 and Bowtie2 v.2.4.2 (40, 50). Then, contigs were clustered into bins with MetaBAT2 (51) and dereplicated with dRep v3.2.2 (10% maximum contamination, 50% minimum completeness) (22). Assembly quality statistics were measured using CheckM v1.1.11 (21) and are available in Table S2.

## Functional annotation

Protein coding sequences were predicted from contigs using Prodigal v2.6.3 (52). To identify the metabolic functions associated with the microbial population of each sample, protein sequences were aligned to the KEGG Orthology database using HMMER v3.1b2 (hmmsearch) (53) filtering for hits with E-value <0.01 and coverage ≥75%.

## Strain profiling

Population-level diversity analyses were performed using InStrain (23) to investigate strain replacements across time points in individuals from UN and from the BIO-ML

cohort from UI. InStrain calculates the ANI between reads and the genomes they map to. We joined, indexed, and aligned dereplicated genomes from each individual and ran InStrain using the profile and compare function, to determine differences between genomes assembled from each individual. InStrain outputs SNP results and calculations for conANI and popANI. InStrain defines conANI as the average similarity between two sequences when considering SNPs that modify the consensus base (major allele) in either sequence. PopANI refers to the average identity between a pair of sequences when taking into account only SNPs that modify more than one allele. Here, we considered genomes to be identical when conANI ≥0.99999 (23). Pairs of genomes with conANI <0.99999 and >0.97 were regarded as distinct strains of the same species, and are referred to as "strain replacements." Pairs with popANI <0.99999 and >0.97 are referred to as "population replacements."

We used a threshold of <0.99999 (23) conANI between genome pairs to identify "strain replacements" and <0.99999 popANI between genome pairs to identify "population replacements." We included genome pairs that were compared along at least 25% of their length. Taxonomic classification of each genome was carried out using GTDB-Tk v.1.0.2 (54) against the Genome Taxonomy Database Release 89.

Further SNP analyses were performed on data obtained through InStrain (26) by applying the "profile" method. SNP calling was based on mapping metagenomic reads to assembled genomes with parameters filtering for a minimum mapping quality (MAPQ) score of 20 and minimum coverage of five reads per site. Next, we employed an approach introduced by Garud et al. (15), in that we selected a major allele frequency threshold (in this case $f \geq 0.7$) to ensure accurate representations of strain replacements. This method relies on the assumption that metagenomic reads will inform the allele distribution of the most abundant genotypes within the sampled community. Therefore, establishing a major allele frequency threshold and observing its frequency over time allows us to infer the ecological dynamics of strains within the host gut. We selected genes that harbored non-synonymous SNPs and met the established quality requirements to perform functional annotations using the KEGG Orthology database as described above.

## ACKNOWLEDGMENTS

We thank the urban Amazonian individuals who participated in this study.

We also thank Conselho Nacional do Desenvolvimento Científico e Tecnológico (CNPq), Coordenação de Aperfeiçoamento de Pessoal de Nível Superior (CAPES), Fundação Amazônia de Amparo a Estudos e Pesquisas (FAPESPA/ Brazil), and Pró-Reitoria de Pesquisa (PROPESP) of Universidade Federal do Pará (UFPA) for the received grants.

The funders had no role in study design, data collection, and interpretation, or the decision to submit the work for publication.

A.P.S., E.J.A., M.G., M.P., and A.R.D.S. designed this study. A.P.S. performed sample collection, metadata collection, and DNA extraction. A.V. performed library preparation and metagenomic sequencing. A.P.S., A.-N.Z., and M.G. performed computational work and analyses. A.P.S. wrote the manuscript, which was improved by all other authors.

## AUTHOR AFFILIATIONS

[1]Genetics and Molecular Biology Program, Universidade Federal do Pará, Belém, Pará, Brazil

[2]Institute of Clinical Molecular Biology, Christian-Albrecht University of Kiel, Kiel, Germany

[3]Schleswig-Holstein University Clinic, Kiel, Germany

[4]Instituto Tecnológico Vale, Belém, Pará, Brazil

[5]Department of Biological Engineering, Massachusetts Institute of Technology, Cambridge, Massachusetts, USA

[6]Center for Microbiome Informatics and Therapeutics, Massachusetts Institute of Technology, Cambridge, Massachusetts, USA

[7]Institute of Experimental Medicine, Christian-Albrecht University of Kiel, Kiel, Germany

[8]The Broad Institute of MIT and Harvard, Cambridge, Massachusetts, USA

[9]The Global Microbiome Conservancy, Massachusetts Institute of Technology, Cambridge, Massachusetts, USA

[10]Center for Oncology Research, Universidade Federal do Pará, Belém, Pará, Brazil

## AUTHOR ORCIDs

Ana Paula Schaan  http://orcid.org/0000-0001-6182-9961
Eric J. Alm  http://orcid.org/0000-0001-8294-9364
Mathieu Groussin  http://orcid.org/0000-0002-0942-7217
Ândrea Ribeiro-dos-Santos  http://orcid.org/0000-0001-7001-1483

## FUNDING

| Funder | Grant(s) | Author(s) |
|---|---|---|
| Coordenação de Aperfeiçoamento de Pessoal de Nível Superior (CAPES) | 88882.160155/2017-01 | Ana Paula Schaan |
| Conselho Nacional de Desenvolvimento Científico e Tecnológico (CNPq) | 304413/2015-1 | Ândrea Ribeiro-dos-Santos |

## AUTHOR CONTRIBUTIONS

Ana Paula Schaan, Conceptualization, Formal analysis, Investigation, Methodology, Visualization, Writing – original draft, Writing – review and editing | Amanda Vidal, Methodology | An-Ni Zhang, Formal analysis | Mathilde Poyet, Conceptualization, Investigation, Supervision, Writing – review and editing | Eric J. Alm, Conceptualization, Supervision | Mathieu Groussin, Conceptualization, Formal analysis, Supervision, Visualization, Writing – review and editing, Resources | Ândrea Ribeiro-dos-Santos, Conceptualization, Funding acquisition, Project administration, Supervision, Writing – review and editing, Resources

## DATA AVAILABILITY

The metagenomics data sets generated and analyzed during the current study are available in the European Nucleotide Archive repository under project number PRJEB58722 (current study) and PRJEB27517 (Tanzania) and NCBI under BioProject PRJNA544527 (United States, BIO-ML). The STORMS checklist can be found at https://doi.org/10.6084/m9.figshare.23653743.v1. Scripts and command lines used to analyze the sequencing and genomic data are available at
https://github.com/anaschaan/amazonMicrobiomeDynamics.

## ETHICS APPROVAL

Written informed consent was obtained from all individuals. Ethics approvals were granted by the Institutional Ethics Review Board of Universidade Federal do Pará, under protocol number 2.686.839. This study was carried out according to the ethical principles established by the Declaration of Helsinki.

## ADDITIONAL FILES

The following material is available online.

### Supplemental Material

**Supplemental Figures (mSystems00707-23-s0001.pdf).** Figures S1-S7.
**Supplemental Tables (mSystems00707-23-s0002.xlsx).** Tables S1-S5.

Open Peer Review

**PEER REVIEW HISTORY (review-history.pdf).** An accounting of the reviewer comments and feedback.

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
