## [Reviewer comments · mSystems]

Temporal dynamics of gut microbiomes in non-industrialized urban Amazonia

Ana Paula Schaan, Amanda Vidal, AnNi Zhang, Mathilde Poyet, Eric Alm, Mathieu Groussin, and Ândrea Ribeiro-dos-Santos

Corresponding Author(s): Ândrea Ribeiro-dos-Santos, Universidade Federal do Pará

Review Timeline:

Submission Date:	July 10, 2023
Editorial Decision:	September 19, 2023
Revision Received:	November 21, 2023
Editorial Decision:	December 16, 2023
Revision Received:	January 17, 2024
Accepted:	January 18, 2024

Editor: Daniel Garrido

Reviewer(s): Disclosure of reviewer identity is with reference to reviewer comments included in decision letter(s). The following individuals involved in review of your submission have agreed to reveal their identity: Jinxin Liu (Reviewer #1); Susheel Bhanu Busi (Reviewer #2)

Transaction Report:

DOI: <https://doi.org/10.1128/msystems.00707-23>

September 19, 2023

Dr. Ândrea Ribeiro-dos-Santos
Universidade Federal do Pará
Rua Augusto Corrêa nº01, Campus do Guamá. Laboratório de Genética Humana e Médica
Belém, Pará 66075-970
Brazil

Re: mSystems00707-23 (Temporal dynamics of gut microbiomes in non-industrialized urban Amazonia)

Dear Dr. Ândrea Ribeiro-dos-Santos:

Thank you for submitting your manuscript to mSystems. We have completed our review and I am pleased to inform you that, in principle, we expect to accept it for publication in mSystems. However, acceptance will not be final until you have adequately addressed the reviewer comments.

While reviewers found the article of interest and were convinced with the conclusions, there are still major issues and questions that need to be resolved.

Preparing Revision Guidelines

Please return the manuscript within 60 days; if you cannot complete the modification within this time period, please contact me. If you do not wish to modify the manuscript and prefer to submit it to another journal, please notify me of your decision immediately so that the manuscript may be formally withdrawn from consideration by mSystems.

Sincerely,

Daniel Garrido

Editor, mSystems

Journals Department
Reviewer comments:

Reviewer #1 (Comments for the Author):

Summary

The authors analyzed the gut metagenomes of urban non-industrialized populations in the Amazon region, focusing on the seasonal dynamics and temporal stability of gut microbiota. They revealed the diversity of gut microbial communities among populations with different industrialization degrees, highlighting the lower stability of the gut microbiome in non-industrialized individuals. Through the long-term tracking of the same individuals' gut genomic profiles, they discovered a higher frequency of strain replacements in the non-industrialized population, and a majority of these non-synonymous variants were associated with genes related to the host's dietary polysaccharide metabolism.

Major comment:

1. Overall the manuscript is clear and well-written, and the topic covered is important and interesting; however, the representativeness of this study is limited (i.e., a small set of samples with only two-time points), particularly given the relatively high individual variations within the non-industrialized population. Additionally, the longitudinal study insufficiently accounted for potential individual factors influencing the gut microbiota, such as lifestyle, diet, and health status; when dealing with a small sample size, these factors could significantly influence the primary outcomes.
2. The discussion within the manuscript appears to be somewhat limited, primarily describing observed phenomena without delving into the underlying biological mechanisms. For instance, while the authors note that a significant proportion of non-synonymous SNP sites are situated within genes associated with host dietary polysaccharide metabolism, they do not provide a thorough exploration of the biological implications of this finding.
3. The manuscript lacks a comprehensive explanation for picking specific time points, especially different seasons, to study the temporal dynamics. The authors should provide additional context regarding whether the population in the studied region experiences substantial lifestyle changes due to seasonal variations. This clarification would help elucidate the potential impact of seasonal factors on the gut microbiome composition and provide a more informed understanding of the study's design and implications.
4. The figure labels throughout the manuscript appear disorganized, which could impede readers' understanding of the content. To enhance clarity and comprehension, the author should consider utilizing a uniform set of axis labels for all figures.
5. As acknowledged by the authors, the substantial absence of longitudinal samples from Tanzania significantly constrains the analysis of the data and the precision of the outcomes, particularly when attempting to compare distinct seasons. Also, are these samples from three different cohorts really comparable? People with distinct host genetics live in very different environments; that is pretty much everything is different.

Minor comment

1. Lines 117-125: The author's account establishes a consistent time gap between the longitudinal sample collections in the UI and UN regions, while the precise months during which the samples were collected remain unspecified. The author should clarify the temporal scope of the initial and subsequent sample collections in the UI region.
2. Lines 122-123: Once acronyms are employed, the entire text should be represented using acronyms. Please review the whole manuscript accordingly.
3. Lines 128-129: The limited number of longitudinal samples from Tanzanian individuals has impacted the assessment of seasonal dynamics in the gut microbiome of the Tanzanian population.
4. Lines 191-197: Please verify the correct placement of figures within the manuscript.
5. Lines 206-215 : Why only the Hadza hunter-gatherers in Tanzania exhibit temporal variations? It should be addressed in the discussion, along with a potential explanation for the biological mechanisms underlying this phenomenon.
6. Figure 2C-D : Adjust the position of the text in the figure to avoid overlapping, and standardize the abbreviations in the diagram, such as United States should be as UI, not USA.
7. Figure 3A: Please carefully review the axis labels of all figures in the manuscript to ensure their accuracy.
8. Figure S4 : Please rearrange the x-axis labels in the figure to match the order used in the previous figure.

Reviewer #2 (Comments for the Author):

General comments:

This paper investigates the temporal dynamics of gut microbiomes in a non-industrialized urban population of the Brazilian Amazon. The study used metagenomic sequencing to characterize the gut microbiomes of 24 individuals and compared the results with available datasets from individuals from divergent lifestyles. The study found that gut microbiome composition and diversity have greater variability over time among non-industrialized individuals when compared to industrialized counterparts. The study also found that tracking microbial population dynamics is important to understand how the microbiome will adapt to transitions. The results suggest that the stability of gut microbiomes is influenced by levels of industrialization. The study concludes that further research is needed to investigate the causal relationships between changes in the gut microbiome and health outcomes.

1. L101: though the authors claim that GM signatures are indicative of urbanisation transition, it is unclear whether the similarities within individuals from the US and Cameroon to that of those in urban Brazil are a function of urbanisation. Please comment on how and why this is urbanisation indeed. Alternatively, could it be that a majority of these taxa have evolved closely with humans and have been fixed in the populations for ages and are not necessarily a sign of urbanisation?
2. L117-125: Whilst the sample design is understandable, the sample collections methods could easily have induced the observed differences in the study. The authors should comment and elaborate on this aspect of the study.
3. L123: The dietary differences between samples collected in Brazil and those from the UI/BIO-ML collection are likely high. The authors have indeed analysed the dietary records of the UN cohort, but their difference with respect to the UI group are not highlighted. Could it be that differences in these dietary preferences are driving the community level differences? Key case-in-point would be the high levels of acai consumption, may indeed have system effects that are manifested via the GM (see: <https://pubmed.ncbi.nlm.nih.gov/33621858/>).
4. L117-125: Why and how did the authors chose the described number of samples for the study? 20+ samples doesn't generally cover the varied diversity of human subjects, so was there any power analyses done? If yes, what metrics were used for the power analyses. If not, it would be prudent to do this analyses and indicate within the manuscript, the power achieved with this modest sample size.
5. L178: How does the ICC analyses compare with generalised linear models? What is the correction factor used in this analyses for the multivariate testing? Please elaborate in the methods. If possible, provide a comparison of the ICC with GLM for the top 10 taxa from both analyses to identify whether ICC robustly captures the multivariate nature of the data.
6. L271: Fig. 4 should be faceted to indicate those that are unique to UN and UI, highlighting overlaps as well in the process. An upset plot would be a better choice for this over a venn diagram.
7. L366: The authors highlight the notion that the SNPs across timepoints show metabolic pathways that are involved in microbial utilisation of host-derived carbohydrates. While it is exciting to see CAZymes being major players, it is not clear which "other" genes also had genes. While it is within the realm of the storyline to focus on these particular genes, this reviewer would like to see the full breakdown of all the genes and their respective functions that demonstrate non-synonymous SNPs across timepoints.
8. L485: What was the rationale behind removed taxa not found in at least 10% of the samples? Was this in any way tested, such as observing diversity metrics across groups at different cutoffs? Why and how was 10% arrived at?
9. L568: While I commend the authors for providing the data for the study, including the STORMS checklist, it is disheartening to see that the "CODE" used to analyse the data in the study has not been provided. It is this reviewer's belief that all code should be made publicly available even at the review stage, in a manner where the study is reproducible.

Dear Dr. Garrido,

Thank you for the opportunity to submit a revision of our manuscript “**Temporal dynamics of gut microbiomes in non-industrialized urban Amazonia**” (mSystems00707-23) for publication at *mSystems*. We appreciate the insightful feedback and the time and attention that you and the reviewers dedicated to reading and improving our paper.

Please find a point-by-point response to the reviewer comments below and a Marked-Up Manuscript file uploaded to the submission system as well, where modifications to the Manuscript text have been highlighted.

Sincerely,

Ândrea Ribeiro-dos-Santos

Ana Schaan

Universidade Federal do Pará
Rua Augusto Corrêa 1
Belém, Pará, Brazil 66075-110
E-mail: akelyufpa@gmail.com

Reviewer #1

Major comment:

1. Overall the manuscript is clear and well-written, and the topic covered is important and interesting; however, the representativeness of this study is limited (i.e., a small set of samples with only two-time points), particularly given the relatively high individual variations within the non-industrialized population. Additionally, the longitudinal study insufficiently accounted for potential individual factors influencing the gut microbiota, such as lifestyle, diet, and health status; when dealing with a small sample size, these factors could significantly influence the primary outcomes.

Response: Thank you for your review and for this thoughtful comment. We appreciate your concern about the sample size in our study. The decision to use this small sample was driven by resource and budget constraints on our end. We acknowledge that this limitation could impact the generalizability of our findings, and therefore it's important to interpret our results considering this constraint. We have included a statement to explicitly discuss this in lines 389-391. We hope for this study to be a catalyzer for future research at our institution that would allow us to scale up our project, increase cohort size and lead to additional discoveries.

We did, however, account for individual factors such as lifestyle, diet, and health status. As described in the Methods section under 'Metadata collection' (Lines 460-467), this information was collected and documented. In our results, we did not see a significant impact of these individual factors on gut microbiome profiles.

2. The discussion within the manuscript appears to be somewhat limited, primarily describing observed phenomena without delving into the underlying biological mechanisms. For instance, while the authors note that a significant proportion of non-synonymous SNP sites are situated within genes associated with host dietary polysaccharide metabolism, they do not provide a thorough exploration of the biological implications of this finding.

Response: We added an additional paragraph within the Discussion section, Lines 373-379, to better discuss the biological implications of non-synonymous mutations on microbial genes responsible for host-derived carbohydrates in the gut microbiome. We hope this expands on our conclusions and clarifies what potential effects this metabolism may have on the host.

3. The manuscript lacks a comprehensive explanation for picking specific time points, especially different seasons, to study the temporal dynamics. The authors should provide additional context regarding whether the population in the studied region experiences substantial lifestyle changes due to seasonal variations. This clarification would help elucidate the potential impact of seasonal factors on the gut microbiome composition and provide a more informed understanding of the study's design and implications.

Response: We fully agree with this suggestion. In selecting our study's time points, we strategically focused on two distinct seasons in the Belém region, mirroring the rainforest seasons experienced by the Hadza foragers of Tanzania. This decision allows for meaningful comparisons with existing data from the Hadza population. Additionally, the Amazon's seasonality could impact the availability of specific food groups, potentially influencing the dietary habits of the local population. This aspect could be particularly interesting in urban Amazonia, where it remains unclear whether substantial dietary changes occur in response to seasonal variations. Furthermore, the increase in rainfall during the Wet season may alter the dynamics of human-environment interactions, given the evolving urban landscape in Belém. Increased flooding (which is frequent in the Wet season), for instance, can impact water sources and contribute to bacterial contamination, thereby influencing the composition of the gut microbiome. In the 'Introduction' section, Lines 104-109, we have added a paragraph to explain our reasoning behind focusing on two distinct rainforest time points.

4. The figure labels throughout the manuscript appear disorganized, which could impede readers' understanding of the content. To enhance clarity and comprehension, the author should consider utilizing a uniform set of axis labels for all figures.

Response: We appreciate your comment and agree with this suggestion. All Figures have been modified to include acronyms RN, UN, and UI to improve clarity and understanding.

5. As acknowledged by the authors, the substantial absence of longitudinal samples from Tanzania significantly constrains the analysis of the data and the precision of the outcomes, particularly when attempting to compare distinct seasons. Also, are these samples from three different cohorts really comparable? People with distinct host genetics live in very different environments; that is pretty much everything is different.

Response: We acknowledge the inherent differences in host genetics and environments across the studied cohorts. However, our study design aimed to capture the broad spectrum of gut microbiome urbanization/industrialization, and our analysis is centered around understanding trends in microbial dynamics and stability over time rather than making direct comparisons of microbial composition across these populations.

Minor comment

1. Lines 117-125: The author's account establishes a consistent time gap between the longitudinal sample collections in the UI and UN regions, while the precise months during which the samples were collected remain unspecified. The author should clarify the temporal scope of the initial and subsequent sample collections in the UI region.

Response: Thank you for bringing this to our attention. We took this into consideration in our analyses and report of results but had not yet included it within the manuscript. Given this is an urban and industrialized environment, and therefore has consistent food availability year-round as well as higher amounts of infrastructure, we did not expect to see fluctuations in gut microbiome profiles across seasons that could potentially be driven by modifications in host-environment interactions. Nevertheless, it is important to note that samples from UI were not collected consistently regarding seasonal differences, although time gaps were consistent.

Due to this, in the alpha diversity analysis of Figure 1B (where we specifically aim to uncover whether seasonality affects microbiome diversity), we included only UI individuals whose T1 sample was collected in either Winter/Fall (i.e. colder periods) and T2 in either Spring/Summer (i.e. warmer periods). We have now included this information within the Manuscript in Lines 454-458. Further, precise sample collection dates are specified within Table S1 for all three cohorts.

2. Lines 122-123: Once acronyms are employed, the entire text should be represented using acronyms. Please review the whole manuscript accordingly.

Response: Thank you for bringing this to our attention. Words with acronyms are now replaced accordingly throughout the entire Manuscript.

3. Lines 128-129: The limited number of longitudinal samples from Tanzanian individuals has impacted the assessment of seasonal dynamics in the gut microbiome of the Tanzanian population.

Response: We fully agree that the lack of representation in longitudinal sampling within the RN cohort limits the generalizability of our findings. We were careful to discuss this as a main limitation of our study in the Discussion sections, Lines 385-389.

4. Lines 191-197: Please verify the correct placement of figures within the manuscript.

Response: Thank you for pointing this out. We revised the entire manuscript and adjusted the mention to the Figures and their respective panels accordingly.

5. Lines 206-215 : Why only the Hadza hunter-gatherers in Tanzania exhibit temporal variations? It should be addressed in the discussion, along with a potential explanation for the biological mechanisms underlying this phenomenon.

Response: We agree that this is an important finding that should be further discussed. These differences in functional profiles among the Tanzania (RN) population have been reported in the original study by Smits et al (2017) and is argued to be driven by the fluctuations in dietary patterns across seasons. These dietary differences lead to the selection of specific taxa specialized in the degradation of plant or animal carbohydrates, for instance. Variability in the metabolic substrates are thus suggested to cause differences in gut metabolic profiles. However, given that this cohort has limited same-individual longitudinal sampling, the observed metabolic profiles could also be due to individual-level differences. Because of this caveat, we did not further emphasize or delve into this finding throughout our discussion. Nevertheless, we have included this information in the Manuscript in Lines 327-332.

6. Figure 2C-D : Adjust the position of the text in the figure to avoid overlapping, and standardize the abbreviations in the diagram, such as United States should be as UI, not USA.

Response: We have reviewed this figure to avoid overlapping of species names. Abbreviations have been standardized to UN, UI, RN.

7. Figure 3A: Please carefully review the axis labels of all figures in the manuscript to ensure their accuracy.

Response: We have reviewed and modified the axis labels in all figures to ensure consistency in populations names (UN, UI, RN).

8. Figure S4 : Please rearrange the x-axis labels in the figure to match the order used in the previous figure.

Response: We have corrected Figure S4 to match the order used in the previous figure.

Reviewer #2 (Comments for the Author):

General comments:

This paper investigates the temporal dynamics of gut microbiomes in a non-industrialized urban population of the Brazilian Amazon. The study used metagenomic sequencing to characterize the gut microbiomes of 24 individuals and compared the results with available datasets from individuals from divergent lifestyles. The study found that gut microbiome composition and diversity have greater variability over time among non-industrialized individuals when compared to industrialized counterparts. The study also found that tracking microbial population dynamics is important to understand how the microbiome will adapt to transitions. The results suggest that the stability of gut microbiomes is influenced by levels of industrialization. The study concludes that further research is needed to investigate the causal relationships between changes in the gut microbiome and health outcomes.

1. L101: though the authors claim that GM signatures are indicative of urbanisation transition, it is unclear whether the similarities within individuals from the US and Cameroon to that of those in urban Brazil are a function of urbanisation. Please comment on how and why this is urbanisation indeed. Alternatively, could it be that a majority of these taxa have evolved closely with humans and have been fixed in the populations for ages and are not necessarily a sign of urbanisation?

Response: Firstly, thank you for the time and attention you have dedicated to reading our Manuscript and for providing feedback.

As you stated, in the Introduction section we mention there is an urbanization gradient in the Brazilian Amazon. This indication stems from our previous study (Schaan et al., 2021) in which we investigated gut microbiomes from populations living across diverse urbanization levels in the Brazilian Amazon region. Across these individuals, we saw a gradient of changes consistent with what is observed among industrialized/urbanized individuals in the United States. A similar study was carried out in Cameroon, where a clear urbanization shift is seen in the gut microbiome of individuals that are transitioning lifestyles. To better contextualize this argument, we have included a short clarification in Line 101.

Given that host genetics is consistent among each of these populations, lifestyle changes driven by urbanization are likely to be the cause of such compositional shifts in the gut microbiome. We hope this clarifies the origin of such statement within the Manuscript.

2. L117-125: Whilst the sample design is understandable, the sample collections methods could easily have induced the observed differences in the study. The authors should comment and elaborate on this aspect of the study.

Response: We fully agree with that differences in sample collection and other steps in sample processing can lead to biases in studies that aim to incorporate data from distinct sources. This is also a matter of concern to us, and we have addressed this issue as a limitation in our Discussion section, Line 385-393.

3. L123: The dietary differences between samples collected in Brazil and those from the UI/BIO-ML collection are likely high. The authors have indeed analysed the dietary records of the UN cohort, but their difference with respect to the UI group are not highlighted. Could it be that differences in these dietary preferences are driving the community level differences? Key case-in-point would be the high levels of acai consumption, may indeed have system effects that are manifested via the GM (see: <https://pubmed.ncbi.nlm.nih.gov/33621858/>).

Response: We appreciate the attention to the potential dietary differences between the UI and UN cohorts. Indeed, differences in gut microbiome compositions are highly driven by dietary patterns, which play a crucial role in characterizing one's lifestyle. Along with diet, we also consider that antibiotic intake, sanitation measures, and subsistence strategies can be highly diverse across lifestyles and may play a role in shaping the gut microbiome. Therefore, yes, it is likely that dietary patterns are one of the main drivers of community level differences between UN and UI cohorts.

Regarding the specific mention of açaí consumption, we did observe an increase in its intake in T2. However, it is noteworthy that in T1, where açaí consumption is comparatively lower (almost zero for most individuals), the microbiome compositions between UN and UI still exhibit significant divergence. This suggests that while açaí consumption may contribute to some level of microbiome variations, it is not the sole driving force behind the observed differences in community composition.

4. L117-125: Why and how did the authors chose the described number of samples for the study? 20+ samples doesn't generally cover the varied diversity of human subjects, so was there any power analyses done? If yes, what metrics were used for the power analyses. If not, it would be prudent to do this analyses and indicate within the manuscript, the power achieved with this modest sample size.

Response: We appreciate the concern with our sample sizes, and we agree this is an important limitation. The decision to use this small sample was driven by resource and budget constraints on our end. A power analysis showed that our current sample size provides a power of 0.4, using $\alpha = 0.05$, Cohen's $d = 0.5$, and sample size = 24. We have now included this information in the manuscript, in Lines 389-391. We acknowledge that this limitation could impact the generalizability of our findings, and therefore it's important to interpret our results considering this constraint. We hope for this study to be a catalyzer for future research at our institution that would allow us to scale up our project, increase cohort size and lead to additional discoveries.

5. L178: How does the ICC analyses compare with generalised linear models? What is the correction factor used in this analyses for the multivariate testing? Please elaborate in the methods. If possible, provide a comparison of the ICC with GLM for the top 10 taxa from both analyses to identify whether ICC robustly captures the multivariate nature of the data.

Response: Thank you for this comment, and we apologize for the confusion. The calculation of ICC incorporates GLM as its first step, when we fit the abundance data in a linear mixed-effects

model that includes partitioning the variance components and accounts for random effects (in this case, individuals). Subsequently, ICC is calculated as the ratio of the variance due to the random variables and the total variance. We used Bonferroni correction to correct for multiple testing, with p-value < 2e-16. In the manuscript this has been clarified in lines 190 & 507-508.

6. L271: Fig. 4 should be faceted to indicate those that are unique to UN and UI, highlighting overlaps as well in the process. An upset plot would be a better choice for this over a venn diagram.

Response: The goal of Figure 4 is to show the number of shared strains between and within individuals, demonstrating that individuals share a higher number of strains with themselves across time than to others. We agree and appreciate your suggestion, and we have indicated each population by the colored points in Fig. 4B and 4C.

7. L366: The authors highlight the notion that the SNPs across timepoints show metabolic pathways that are involved in microbial utilisation of host-derived carbohydrates. While it is exciting to see CAZymes being major players, it is not clear which "other" genes also had genes. While it is within the realm of the storyline to focus on these particular genes, this reviewer would like to see the full breakdown of all the genes and their respective functions that demonstrate non-synonymous SNPs across timepoints.

Response: Thank you for this comment. To further clarify our results in the SNP analysis, we included a new Table S5 that contains the full breakdown of gene functions and taxonomy for all the genes that harbored nonsynonymous SNPs between timepoints across both populations. This is mentioned in the Manuscript in Line 288.

8. L485: What was the rationale behind removed taxa not found in at least 10% of the samples? Was this in any way tested, such as observing diversity metrics across groups at different cutoffs? Why and how was 10% arrived at?

Response: Thank you, this is a very relevant question. Yes, cutoffs were tested at sample representation frequencies of 5%, 10% and 20%. We found that 10% was the best balance between capturing biologically relevant taxa and excluding noise, aligning with established practices in the field and maintaining statistical robustness. At genus level, 10% represents 440 taxa. Increasing this cutoff to 20%, for instance, led to a big loss of diversity, particularly from species in gut microbiomes of Tanzania.

9. L568: While I commend the authors for providing the data for the study, including the STORMS checklist, it is disheartening to see that the "CODE" used to analyse the data in the study has not been provided. It is this reviewer's belief that all code should be made publicly available even at the review stage, in a manner where the study is reproducible.

Response: We fully agree that availability of analysis code is important to reinforce open access science and ensure reproducibility. We have made our script available at

<https://github.com/anaschaan/amazonMicrobiomeDynamics>. This statement is included in “Data availability”, Lines 578-579.

Re: mSystems00707-23R1 (Temporal dynamics of gut microbiomes in non-industrialized urban Amazonia)

Dear Dr. Ândrea Ribeiro-dos-Santos:

Revision Guidelines

Sincerely,
Daniel Garrido
Editor
mSystems

Reviewer #1 (Comments for the Author):

all my comments have been sufficiently addressed

Reviewer #2 (Comments for the Author):

Thank you for making the necessary revisions. Please include your response to question 3 regarding the Acai consumption in the discussion.

Dear Dr. Garrido,

Thank you for the opportunity to re-submit a revision of our manuscript “**Temporal dynamics of gut microbiomes in non-industrialized urban Amazonia**” (mSystems00707-23) for publication at *mSystems*. We appreciate the time and attention that you dedicated to reading our paper.

Below you will find responses to the comments left by the reviewers in the second round of reviews, as well as Marked-Up Manuscript file uploaded to the submission system, where modifications to the Manuscript text have been highlighted.

Sincerely,

Ândrea Ribeiro-dos-Santos Ana Schaan

Universidade Federal do Pará Rua Augusto Corrêa 1
Belém, Pará, Brazil 66075-110 E-mail: akelyufpa@gmail.com

Reviewer #1 (Comments for the Author):

all my comments have been sufficiently addressed

Response: Thank you for the insightful feedback and the time dedicated to improving our manuscript.

Reviewer #2 (Comments for the Author):

Thank you for making the necessary revisions. Please include your response to question 3 regarding the Acai consumption in the discussion.

Response: Thank you for the insightful feedback and the time dedicated to improving our manuscript. We added the discussion regarding açai consumption in the Discussion section, Lines 303-307.

Ândrea Ribeiro-dos-Santos

Re: mSystems00707-23R2 (Temporal dynamics of gut microbiomes in non-industrialized urban Amazonia)

Dear Dr. Ândrea Ribeiro-dos-Santos:

Your manuscript has been accepted, and I am forwarding it to the ASM production staff for publication. Your paper will first be checked to make sure all elements meet the technical requirements. ASM staff will contact you if anything needs to be revised before copyediting and production can begin. Otherwise, you will be notified when your proofs are ready to be viewed.

Featured Image Submissions: If you would like to submit a potential Featured Image, please email a file and a short legend to msystems@asmusa.org. Please note that we can only consider images that (i) the authors created or own and (ii) have not been previously published. By submitting, you agree that the image can be used under the same terms as the published article. Image File requirements: TIF/EPS, 7.5 inches wide by 8.25 inches tall (at least 2,250 pixels wide by 2,475 pixels tall), minimum 300 dpi resolution (600 dpi preferred), RGB, and no figure elements, e.g., arrows or panel labels. The legend should be a short description of the image, 1-2 sentences recommended.

Sincerely,
Daniel Garrido
Editor